

# Gravity Waves excited during a Minor Sudden Stratospheric Warming

Andreas Dörnbrack[1], Sonja Gisinger[1], Natalie Kaifler[1], Tanja Portele[1], Martina Bramberger[1], Markus Rapp[1,2], Michael Gerding[3], Jens Faber[3], Nedjeljka Žagar[4], and Damjan Jelić[4]

[1] Institut für Physik der Atmosphäre, DLR Oberpfaffenhofen, Oberpfaffenhofen, Germany
[2] Meteorologisches Institut München, Ludwig-Maximilians-Universität München, Munich, Germany
[3] Leibniz Institute of Atmospheric Physics at the University of Rostock, Kühlungsborn, Germany
[4] University of Ljubljana, Faculty of Mathematics and Physics, Department of Physics, Ljubljana, Slovenia

*Correspondence to*: Andreas Dörnbrack (andreas.doernbrack@dlr.de)

**Abstract.** An exceptionally deep upper-air sounding launched from Kiruna airport (67.82°N, 20.33°E) on 30 January 2016 stimulated the current investigation of internal gravity waves excited during a minor sudden stratospheric warming (SSW) in
the Arctic winter 2015/16. The analysis of the radiosonde profile revealed large kinetic and potential energies in the upper stratosphere without any simultaneous enhancement of upper tropospheric and lower stratospheric values. Upward propagating inertia-gravity waves in the upper stratosphere and downward propagating modes in the lower stratosphere indicated a region of gravity wave generation in the stratosphere. Two-dimensional wavelet analysis was applied to vertical time series of temperature fluctuations in order to determine the vertical propagation direction of the stratospheric gravity
waves in one-hourly high-resolution meteorological analyses and short-term forecasts. The separation of up- and downward propagating waves provided further evidence for a stratospheric source of gravity waves. The scale-dependent decomposition of the flow into a balanced component and inertia-gravity waves showed that coherent wave packets preferentially occurred at the inner edge of the Arctic polar vortex where a sub-vortex formed during the minor SSW.

## 1. Introduction

Stratospheric gravity waves observed at middle and high latitudes during wintertime conditions are occasionally attributed to spontaneous adjustment related to the polar-night jet (e.g. Sato 2000, Sato et al., 2012). The polar-night jet (PNJ) is a circumpolar stratospheric jet generated by the hibernal cooling at high latitudes. The resulting meridional temperature gradient generates strong westerly winds. Due to the underlying geostrophic or gradient wind balance, the PNJ is predominantly a balanced phenomenon.  However, in the northern hemisphere the stratospheric polar vortex is usually
disturbed by planetary waves leading to different kinds of sudden stratospheric warmings (SSWs, e.g. Charlton and Polvani, 2007, Butler et al., 2015). Departures from the balanced state lead to flow adjustment processes which can radiate as internal



gravity waves from the jet stream (Limpusavan et al., 2011). This source mechanism to generate internal gravity waves is known as spontaneous adjustment (e.g., Plougonven and Zhang, 2014).

Sato (1999, 2000) was among the first who suggested a similarity of the spontaneous adjustment at the tropospheric jet with that occurring in the stratosphere: Her numerical simulations using a gravity-wave resolving global circulation model revealed a "dominance of downward propagation of wave energy around the polar-night jet in the winter hemisphere, suggesting the existence of gravity wave sources in the stratosphere." More recently, Sato et al. (2012) used the gravity wave potential energy $E_p$ as determined from their model simulations to document the longitudinal and latitudinal distribution of gravity waves in the lower stratosphere in the southern winter hemisphere. They found significant downward energy fluxes associated with gravity waves in the lower stratosphere to the south of the Southern Andes. Besides the downward spread of partially reflected mountain waves from the Andes, non-linear processes in the stratosphere were mentioned as likely reasons.

Direct observations of the excitation of stratospheric gravity waves by non-linear processes are rare. Most of the existing studies concentrate on statistical aspects derived from high-vertical resolution radiosondes (e.g., Yoshiki and Sato, 2000; Yoshiki et al., 2004), from ground-based (e.g., Whiteway et al., 1997, Whiteway and Duck, 1999, Khaykin et al., 2015) and space-borne (e.g., Wu and Waters, 1996, Hindley et al., 2015) remote-sensing observations. The observations of Whiteway at Eureka, Nunavut, Canada (80°N) during four Arctic winters showed an increase of stratospheric gravity wave energy in the vicinity of the PNJ. More precisely, the amount of upper stratospheric wave energy was maximum within the PNJ at the edge of the polar vortex, minimum near the vortex center, and intermediate outside the vortex (Whiteway et al., 1997).

Yoshiki and Sato (2000) analyzed radiosonde observations from 33 polar stations over a period of 10 years to investigate gravity waves in the lower stratosphere, inter alia by examining the correlation between the gravity wave intensity (expressed as kinetic energy $E_K$) in the lower stratosphere and the mean wind. For the Arctic, they found a high correlation of $E_K$ with the surface wind whereas $E_K$ correlates with the stratospheric wind in the Antarctic. The dominance of upward propagating gravity waves points to orographic sources in the Arctic whereas the high percentage of downward energy propagation in the lower stratosphere found for winter and spring suggests other gravity wave sources in the Antarctic. Yoshiki and Sato (2000) speculated that one source candidate is likely to be the PNJ. Yoshiki et al. (2004) investigated the temporal variation of the gravity wave energy in the lower stratosphere with respect to the position of the polar vortex by using the equivalent latitude coordinate for the Antarctic station Syowa (69°S, 40°E). In agreement with the results of Whiteway et al. (1997), gravity wave energy is enhanced when the edge of the polar vortex approaches Syowa Station. Surprisingly, they found an especially large enhancement during the breakdown phase of the polar vortex in spring. As they write in their paper: "As it is difficult to explain the energy enhancements only by the variation in horizontal wind and/or the static stability, the enhancements of wave activity at the edge of the polar vortex are likely to contribute to the energy enhancements by wave generation in the stratosphere.".



Hindley et al. (2015) suggested that the distributions of increased stratospheric $E_p$–values in the southern hemisphere eastwards of around 20° E as determined from Global Positioning System radio occultation (GPS-RO) data from the COSMIC satellite constellation might be due to stratospheric sources. Especially, their nearly homogeneous distribution of enhanced $E_p$–values indicates "… a zonally uniform distribution of small amplitude waves from non-orographic mechanisms such as spontaneous adjustment and jet instability around the edge of the stratospheric jet." (p. 7813, Hindley et al., 2015).

In-situ observations in the stratosphere from 20 to 40 km altitude are rare, especially at high latitudes. As in the studies by Yoshiki and Sato (2000) and Yoshiki et al. (2004), most of the published gravity wave analysis relies on operational radiosondes launched once or twice a day. At high latitudes, only few of these radiosondes reach altitudes higher than 30 km in winter as the conventional 300 … 500 g rubber balloons burst in the cold stratosphere. During the METROSI[1] campaign in Northern Scandinavia, 3000 g rubber balloons were used by the LITOS group (Theuerkauf et al, 2011, Haack et al., 2014, Schneider et al., 2017) studying fine-scale turbulence in the upper troposphere/lower stratosphere (UTLS) mainly from Andøya, Norway (69°N, 15.7°E). A common deployment phase of their turbulence sensor took place in Kiruna, Sweden (68°N, 20°E) during GW-LCYCLE 2[2]. Operational forecasts of the integrated forecast system (IFS) of the European Centre for Medium-Range Weather Forecasts (ECMWF) predicted the appearance of wavelike structures in the stratosphere northwest of Kiruna. The close proximity of the predicted waves and the relatively weak stratospheric winds ($\lesssim$ 50 m s$^{-1}$) led to the decision to launch a radiosonde with one of the large 3000 g balloons in the morning of 30 January 2016.

The relatively weak stratospheric winds over Northern Scandinavia were associated with the southward displacement of the Arctic polar vortex during a minor SSW (Matthias et al., 2016, Manney and Lawrence, 2016). The stratosphere in the northern hemispheric winter 2015/2016 was exceptionally cold as the polar vortex was essentially barotropic and centered near the North Pole in early winter months (Matthias et al., 2016). These conditions are usually associated with weak planetary wave activity. Indeed, Matthias et al. (2016) showed that the planetary wave number 1 amplitude was exceptionally small in November/December 2015 compared to 37 years of ERA-Interim and 68 years of NCAR/NCEP reanalysis data. Planetary waves of zonal wave number 1 were amplified during the second half of January 2016 and three consecutive minor SSWs occurred before the final breakdown of the polar vortex at the beginning of March 2016 (Manney and Lawrence, 2016).

This paper presents a case study analyzing a series of radiosonde observations during the first minor SSW end of January 2016. We focus on the analysis of the 3000 g balloon ascent of 30 January 2016 which reached an exceptional altitude of 38.1 km. The paper documents this event by combining and comparing the measurements with numerical weather prediction

---

[1] METROSI: Mesoscale Processes in Troposphere-Stratosphere Interaction

[2] The GW-LCYCLE 2 campaign was a coordinated effort of multiple German institutions to combine ground-based, ballon-borne, airborne and satellite instruments to investigate the life cycle of gravity waves above Scandinavia in January and February 2016.





(NWP) analyses and forecasts. The characteristics and the sources of the observed stratospheric gravity waves are investigated. We show that the characteristics of the stratospheric gravity waves determined from observations and model data suggest a stratospheric source. Section 2 contains information about the data sources and the methods to analyze them. Section 3 reviews the particular meteorological situation and portrays the transition of the stratospheric flow regime over

5 Northern Scandinavia. Section 4 presents the vertical profiles of the deep radiosonde sounding and continues with an analysis of the observed gravity waves. Section 5 investigates the wave properties derived from the IFS data starting with a comparison to the observations, and section 6 concludes the paper.

## 2. Data Sources and Analysis Methods

### *Radiosonde soundings*

Nine consecutive radiosondes were launched from Kiruna airport (67.82°N, 20.33°E) on 29 and 30 January 2016. The radiosondes were the Vaisala model RS41-SG (Vaisala, 2017). The measured horizontal wind components u and v and temperature T have a temporal resolution of one second. Assuming a mean balloon ascent rate of 5 m s$^{-1}$ the atmospheric variables u, v, and T have a vertical resolution of about 5 m. For the wave analysis the radiosonde data were interpolated onto an equidistant vertical grid with 25 m resolution. The gravity wave properties of all nine soundings were analyzed

whereby the wave perturbations $u', v',$ and $T'$ were calculated as differences between the actual quantities and the respective background profiles $\langle u \rangle, \langle v \rangle,$ and $\langle T \rangle$. The background profiles were determined by a second-order polynomial fit of u, v and T in the troposphere and in the stratosphere, respectively. Additionally, a 5 km running mean was removed from the perturbation profiles and added to the background (Lane et al., 2000, Lane et al., 2003). The latter step reduces the arbitrariness of the polynomial fit depending on the particular shape of the measured profiles and avoids outliers in the

perturbation profiles. The specific kinetic and potential energies are calculated according to $E_K = \frac{1}{2}\left(\overline{u'^2} + \overline{v'^2}\right)$ and $E_P = \frac{g^2}{2N^2}\frac{\overline{T'^2}}{\langle T \rangle^2}$, respectively, whereby the overbar denotes averages taken over selected layers in the troposphere or stratosphere. Stokes parameters and rotary spectra are used to describe essential parameters of the gravity waves retrieved from the perturbation wind components $(u', v')$ and from $T'$. The associated techniques are well documented and described, e.g. by Vincent (1984), Eckermann and Vincent (1989), Eckermann (1996), Vincent et al. (1997), and Murphy et al. (2014).

### 25 *Meteorological data from the IFS*

Operational analyses and high-resolution (HRES) forecasts of the ECMWF's IFS are used to provide meteorological data characterizing the ambient atmospheric conditions and the resolved stratospheric gravity waves. The IFS is a global, hydrostatic, semi-implicit, semi-Lagrangian NWP model. Two different IFS cycles are available for January 2016. The operational IFS cycle 41r1 provides fields with a horizontal resolution of about 16 km (T$_L$1279) and 137 vertical model

levels (L137). The IFS cycle 41r2 has a horizontal resolution of about 9 km (T$_{Co}$1279) and the same number of vertical



levels[3] (Hólm et al., 2016). With the cubic spectral truncation used for cycle 41r2 the shortest resolved wave is represented by four rather than two grid points and the octahedral grid is globally more uniform than the previously used reduced Gaussian grid (Malardel and Wedi, 2016). The model top of both IFS cycles is 0.01 hPa. The IFS cycle 41r2 was in its pre-operational mode and products were disseminated among the users. Ehard et al. (2018) showed that both IFS cycles reproduce the temporal evolution of the observed gravity wave potential energy density $E_P$ in the middle stratosphere above Sodankylä, Finland correctly for the months December 2015 to March 2016. Therefore, the IFS cycle 41r2 was selected for the present analysis and profiles of the IFS cycle 41r1 are only shown for comparison in Figure 3.

*Scale-dependent modal decomposition*

The IFS cycle 41r2 analyses of 30 January 2016 00, 06, and 12 UTC are decomposed into inertia-gravity waves (IGWs) and Rossby waves using the 3D normal-mode function decomposition as described by Žagar et al. (2015). The modal decomposition projects the 3D wind and geopotential height fields onto a set of predefined basis functions for a realistic model stratification. The basis functions are derived from an eigenvalue problem (Kasahara 1976, 1978, Kasahara and Puri 1981). The eigenfunctions are solutions to two dispersion relationships which correspond to the vorticity dominated Rossby waves and divergence-dominated IGWs on the sphere (Kasahara 1976). A distinct advantage of the applied decomposition is the 3D orthogonality of the basis functions which enables filtering of any mode of oscillations in physical space. Žagar et al. (2017) applied the modal decomposition to recent ECMWF analyses and discussed the IGW features across the resolved spectrum. Their examples of propagating linear IGWs in the data analyzed at selected time steps showed that the modal decomposition is a useful complementary tool for studying internal gravity waves.

The results of Žagar et al. (2017) suggest that current ECMWF analyses well resolve IGWs in synoptic scales and in large mesoscale features with scales larger than 500 km. For smaller scales, such as studied here, the model spectrum of IGWs deviates from the expected -5/3 slope suggestive of a lack of variability. Our comparison of the model with observations will illustrate some aspects of the missing variability. By filtering out zonal wavenumbers smaller 30, we focus on inertia-gravity waves with horizontal wavelengths shorter than 660 km (at 60°N). The IGWs are evolving on the background flow that is represented by the Rossby waves and this flow component is denoted by BAL (balanced). The balanced component is presented without scale-dependent filtering.

*Two-dimensional Wavelet Analysis*

In addition to the normal-mode decomposition, we analyze IFS cycle 41r2 temperatures in the time frame between 26 January and 1 February 2016 for gravity waves. For this purpose, the one hourly temperature fields of the IFS cycle 41r2 above Kiruna are interpolated on an equidistant grid with 500 m vertical resolution in the altitude range of 12 to 65 km. This

---

[3] The IFS cycle 41r2 became operational on 8 March 2016; see
https://software.ecmwf.int/wiki/display/FCST/Implementation+of+IFS+cycle+41r2 and
http://www.ecmwf.int/en/about/media-centre/news/2016/new-forecast-model-cycle-brings-highest-ever-resolution



dataset might be considered as "virtual" ground-based measurements, e.g. by a Rayleigh lidar, emulating a common type of observations during Arctic field campaigns (e.g. Baumgarten et al., 2015, Hildebrand et al., 2017, Kaifler et al., 2015). Temperature perturbations T′ assigned to gravity waves are determined from the local temperatures relative to an area mean between 65°N, 10°E and 70°N, 30°E. In this way, perturbations with scales larger than 600 … 900 km horizontal wavelength are effectively removed. No vertical or temporal filters were applied. We cross-checked with the methods commonly applied to Rayleigh lidar measurements (Ehard et al., 2015) and found that (i) the problems arising in vertical filtering due to the tropospheric T gradient and the tropopause were circumvented, (ii) mountain wave signatures are sustained compared to temporal filtering methods, and (iii) our results on transient gravity waves in the stratosphere were unaffected. In order to distinguish upward and downward propagating gravity waves, we applied directional two-dimensional Morlet wavelets (Wang and Lu, 2010) to T′ as a function of altitude and time. The wavelet analysis allows for separatation of gravity waves of different vertical wavelengths and phase speeds while preserving temporal and vertical information. Using this method, the occurrence and activity of quasi-stationary as well as transient upward and downward propagating waves is investigated and the vertical wavelength $\lambda_z$ and ground-based vertical phase speed $c_{Pz}$ of dominant gravity waves are estimated. To investigate the evolution of the gravity wave field around 30 January relative spectrograms are determined. They are the differences between the spectrogram computed for selected intervals and the global spectrogram computed over the whole period from 26 January till 1 February 2016. As the latter contains the sum of all contribution from waves, the values of the relative spectrograms are negative. The highest value zero means that the wave packet in question was detected in the selected interval only. Recently, this technique was developed and applied successfully to time series of ground-based Rayleigh lidar profiles and is described and discussed in detail by Kaifler et al. (2015, 2017).

## 3. Meteorological Situation

*Circulation pattern*

Weak stratospheric planetary wave activity caused exceptionally low temperatures inside the Arctic polar vortex in the early winter months 2015/2016 (Matthias et al., 2016, Dörnbrack et al., 2017a). The polar vortex remained cold until the beginning of March 2016 (Manney and Lawrence, 2016). Starting mid-January 2016, the polar vortex became disturbed by planetary waves and three minor SSWs which occurred at the end of January and mid-February 2016. During the January 2016 minor SSW, the center of the polar vortex was displaced from the pole region and shifted southward between Northern Scandinavia and Svalbard. The circumpolar PNJ elongated in the west-east direction above Eurasia leading to regions of strong curvature at the vertices over the Northern Atlantic and over Siberia. Figures 1a, b, and c illustrate the twist of the vortex and its distortion by means of the magnitude of the balanced wind $V_H^{BAL}$ at three stratospheric pressure levels valid at 06 UTC on 30 January 2016.



Near the stratopause at 1 hPa, the PNJ was strongly decelerated to values $V_H^{BAL} < 30$ m s$^{-1}$ in the region south of Greenland (Fig. 1a). Above Europe, the jet core was located south of the Alps and attained maximum winds of more than 120 m s$^{-1}$ at this pressure level. Northern Scandinavia was located in the center of the polar vortex where $V_H^{BAL} \lesssim 20$ m s$^{-1}$. However, north of Scotland and west of Scandinavia, a jet branch separated from the inner edge of the PNJ in the strong curvature region over the Northern Atlantic and a weak rotary circulation formed inside the polar vortex (Fig. 1a). A band of enhanced balanced winds $V_H^{BAL} \approx 40 \ldots 50$ m s$^{-1}$ is also visible at the same location at 5 hPa (Fig. 1b) extending towards Northern Scandinavia. Otherwise, Fig. 1b documents a closed circulation of the polar vortex with $V_H^{BAL} > 80$ m s$^{-1}$ in the elongated parts and weaker winds near the eastern and western vertices. The separated upper stratospheric branch of the PNJ was on top of a region of $V_H^{BAL} \approx 30 \ldots 50$ m s$^{-1}$ in the middle stratosphere at 50 hPa (Fig. 1c). The nearly unidirectional strong southwesterly stratospheric winds over Southern and Northern Scandinavia suggest favorable propagation conditions for gravity waves.

Near the tropopause level at 250 hPa, three jet streaks with maximum $V_H^{BAL} \approx 70$ m s$^{-1}$ stretched across the North Atlantic and the North Sea (Fig. 1d). They led to high tropospheric winds over Scotland and Southern Scandinavia. Northern Scandinavia was north of the baroclinic zone associated with the polar front and influenced by a weak tropopause jet with maximum $V_H^{BAL} \approx 25$ m s$^{-1}$ oriented nearly perpendicular to the mountain range.

*Temporal Evolution above Kiruna*

Figure 2 illustrates the temporal evolution of the horizontal wind $V_H$ (Fig. 2a) and the absolute temperature T (Fig. 2b) over Kiruna in the period from 26 January until 1 February 2016. In general, this period is characterized by a regime transition of the stratospheric flow during the minor warming. The gradual decline of the horizontal wind (Fig. 2a) and the descent of the cold layer (Fig. 2b) in the stratosphere reflect the southward displacement of the polar vortex and the approach of its center. After 30 January 2016, the conditions over Kiruna are marked by light horizontal wind $V_H < 20$ m s$^{-1}$, a warmer stratopause of up to 290 K, and an about 3 … 4 km lower cold stratospheric layer. In this layer, the IFS temperature was about 4 to 8 K warmer than at the beginning of the period. During the regime transition, $V_H$, T, and $\Theta$ display wave-like perturbations in the stratosphere (Fig. 2).

**4. The deep radiosonde sounding of 30 January 2016**

*Radiosounding*

On 29 and 30 January 2016, altogether nine radiosondes were launched from Kiruna airport. Only two radiosondes ascended to altitudes larger than 30 km. For these upper-air soundings 3000 g rubber balloons (TOTEX, TX3000) were employed, the smaller 500 g or 600 g balloons used for the other soundings burst in the cold layer of the polar vortex (see vertical trajectories in Fig. 2b). The minimum temperature measured during these two days was $T_{MIN} \approx 180$ K ($-93$°C) on 29 January



at 08:45 UTC at 25 km altitude (not shown). However, the observed $T_{MIN}$ increased by about 10 K above Kiruna during 24 hours due to the adiabatic descent associated with the southward shift of the polar vortex. Nevertheless, only the large 3000 g balloons could penetrate the cold stratospheric layer without bursting.

On 30 January 2016, the 3000 g rubber balloon was released at 09:08:45 UTC and the ascent lasted until 10:52:17 UTC
reaching an altitude of 38.1 km. During this time, the balloon drifted about 160 km to the northeast. Figure 3a shows the temperature profile as function of altitude. The bow-shaped profile is characterized by a minimum temperature of about 190 K (−83° C) inside the polar vortex between 20 and 28 km altitude. This value is about 45 K lower than the temperature at the tropopause which is marked by the strong increase of static stability at ≈ 8 km altitude (Fig. 3c). Above 28 km altitude the temperature increased by about 55 K and attained similar values at 38.1 km as measured in the troposphere at around 6 km
altitude. The other characteristic of the temperature profile is the existence of wave-like oscillations. They are pronounced between 10 and 20 km and above about 28 km altitude. The amplitude of the fluctuations increases with altitude which is also visible in the profile of the potential temperature Θ (Fig. 3b). The buoyancy frequency as calculated from the Θ-profile shows substantial oscillations with increasing amplitudes in the stratosphere (Fig. 3c). In the upper range of the profile, very small and even negative values were calculated indicating the existence of vertically separated mixing layers.

Overall, the magnitude of the horizontal wind $V_H$ as depicted in Fig. 3d increases nearly linearly from about 5 m s$^{-1}$ at 1 km to about 50 m s$^{-1}$ at 38 km altitude. From 29 to 30 January 2016, the mean horizontal wind in the lower troposphere decreased from 20 m s$^{-1}$ (not shown) to about 10 m s$^{-1}$ (Fig. 3d). The main features of the $V_H$-profile are fluctuations with an apparent vertical wavelength of about $\lambda_z \approx 3 \ldots 5$ km and shorter oscillations with $\lambda_z \approx 1$ km. The amplitude of the longer waves increases with height whereas the shorter waves almost vanish above 28 km altitude. Figure 3e shows the turning of
the wind from southerlies in the troposphere to south-westerlies which dominated the stratospheric flow above Northern Scandinavia on 30 January 2016. The wave-like oscillations of the wind direction reflect the same kind of pattern as found in the $V_H$-profile. The vertical wind as derived from the balloons ascent rate and from the IFS show a growth in amplitude with increasing altitude (Fig. 3f). However, the amplitudes and vertical wavelengths differ due to the absence of high-frequency waves in the IFS. In the following subsection, properties of the observed gravity waves are analyzed.

*Gravity wave analysis*

As listed in Table 1, the vertically averaged kinetic and potential energies $E_K$ and $E_P$ in the troposphere (1 … 8 km) and lower stratosphere (12 … 20 km) of the deep sounding on 30 January 2016 have values of $E_K = 8$ (10) J kg$^{-1}$ and $E_P = 10$ (5) J kg$^{-1}$, respectively. These values are close to the mean over all nine soundings of $E_K = 10$ (9) J kg$^{-1}$ and $E_P = 7$ (4) J kg$^{-1}$ for the troposphere and lower stratosphere. In the middle stratosphere (20 ... 28 km), vertically averaged energies $E_K$ and
$E_P$ increase to values of 23 and 9 J kg$^{-1}$, respectively. Compared to the earlier deep sounding on 29 January 2016 10:42 UTC (reaching only 31 km peak altitude), there is nearly a doubling of tropospheric and stratospheric gravity wave potential




energies whereas the tropospheric kinetic energy is halved (cf. Table 2 with Table 1). The reduction of tropospheric gravity wave kinetic energy goes along with weakening tropospheric winds as mentioned above. High energy values are obtained in the upper stratosphere (30 … 38 km) of $E_K = 79$ J kg$^{-1}$ and $E_P = 25$ J kg$^{-1}$ on 30 January (Table 1). Besides the increase of $E_K$ and $E_P$ in the stratosphere above 20 km altitude, we found a dominance of the kinetic energy in these layers for both deep

soundings of 29 and 30 January 2016 (Tables 1, 2). According to Sato and Yoshiki (2008), $E_K$-values larger than $E_P$ point to inertia-gravity waves with an intrinsic frequency $\Omega$ close to the inertia frequency $f$ as $E_p/E_K = (\Omega^2 - f^2)/(\Omega^2 + f^2)$ based on linear theory, see Gill (1982). Applying this relationship results in an estimate of the scaled intrinsic frequency being $\Omega/f \approx 1.4 … 1.7$ for the stratospheric layers on 30 January. This means, the observed waves have intrinsic frequencies much smaller than the buoyancy frequency $N \gtrsim 0.02$ s$^{-1}$ (Fig. 3c), i.e. this analysis points to essentially low-frequency, hydrostatic

inertia-gravity waves (Fritts and Alexander, 2003).

Stokes analysis was applied to determine essential gravity wave parameters (e. g., Eckermann and Vincent, 1989, Vincent et al., 1997). First, the degree of polarization was determined to be 0.8 between 30 and 38 km altitude for the 30 January 2016 09:08 UTC profile. This value becomes gradually smaller for the lower stratospheric layers (0.7 for 20 … 28 km and 0.3 for 12 … 20 km). Values close to one point to monochromatic waves while values close to zero indicate a random wave field

(Vincent et al., 1997). The middle stratospheric fluctuations determined from the radiosonde profile are thus dominated by coherent gravity waves obeying the linear dispersion relation. The ratio $\Omega/f$ determines the hodographs of u' and v' and can be calculated by Stokes analysis. The small values of $\Omega/f \approx 2.1$ for 30 … 38 km and $\Omega/f \approx 3.2$ for 20 … 28 km support the former finding that the observed waves are most likely low-frequency inertia-gravity waves. On the other hand, the larger ratio of $\Omega/f \approx 12$ derived for the lower stratosphere might point to an influence of different types of gravity waves, see

discussion Section 6.

The vertical wavelength determined from the power spectra of u' and v' is $\lambda_z \approx 4$ km in all altitude layers. The horizontal wavelength determined by using $\Omega/f$ from the Stokes analysis increases with height from $\lambda_H \approx 50$ km for 12 … 20 km altitude to $\lambda_H \approx 220$ km (20 … 28 km), and, finally, to $\lambda_H \approx 330$ km between 30 and 38 km. Applying the Stokes analysis, the horizontal propagation direction turns anti-clockwise from west-northwest in the lower stratosphere to west between 20

… 28 km, to finally west-southwest in the uppermost layer (30 … 38 km). Therefore, the inertia-gravity waves with an estimated intrinsic phase speed $c_p$ between 13 and 15 m s$^{-1}$ propagated against the ambient stratospheric flow. As the magnitude of the intrinsic phase speed $c_p$ is smaller than the ambient wind $V_H$, the gravity waves propagate northeastward with respect to the ground.

Both the rotary spectra and the Stokes analysis can be used to estimate the dominant vertical propagation direction for the

30 deep soundings. For this purpose, the ratio $R^{RS}$ of the power of the upward propagating part of the rotary spectrum, i.e. the positive part of the Fourier spectrum of u'+iv' (Vincent, 1984), to the total power is calculated and listed in Table 1 and 2.





For $R^{RS} > 0.6$ upward wave propagation can be assumed because there is a significant bias towards upward propagation of inertia-gravity wave (Guest et al., 2000). $R^{RS}$ is 0.78, 0.51, 0.62, and 0.76 for the layers 1 … 8 km, 12 … 20 km, 20 … 28 km, and 30 … 38 km altitude, respectively, for the 30 January 2016 09:08 UTC sounding. Thus, except in the layer from 12 to 20 km altitude, the dominant propagation direction is upward. This result is supported by the Stokes analysis which gives downward propagation exclusively in the same layer from 12 to 20 km altitude. The deep radiosonde sounding from 29 January 2016 10:42 UTC clearly indicates downward propagating waves in an extended altitude range between 12 and 28 km (Table 2). Assuming the downward propagating waves are radiated from the same elevated source, the excitation likely occurred shortly before 29 January 2016 ~11 UTC at an altitude above 28 km and its effects are still detected in the lower stratosphere 22 h later.

## 5. Gravity wave analysis of IFS data

*Comparison of radiosonde observations with IFS*

Vertical profiles of different variables from the ECMWF IFS interpolated on the balloon trajectory in space and time are shown in Figure 3. The IFS data used for this comparison are the short-term HRES forecasts of the IFS cycles 41r1 and 41r2 at lead times +9 h, +10 h, and +11 h of the 00 UTC forecast run of 30 January 2016.

The observed and simulated temperature profiles agree qualitatively and quantitatively very well up to an altitude of 28 km. Obviously, the IFS profiles cannot reproduce the fine-scale oscillations found in the temperature sounding. However, the general temperature decrease and the cold stratospheric layer are quantitatively well captured by the model. Higher up, the profile of IFS cycle 41r2 captures the local T-maximum at about 34 km well whereas the coarser resolved cycle 41r1 underestimates the stratospheric fluctuations. Comparing the two different IFS cycles, it becomes clear that the IFS cycle 41r1 has a smaller wave amplitude $\Delta T \approx 1.4$ K whereas the IFS cycle 41r2 has a nearly realistic $\Delta T \approx 5$ K. Deviations between the IFS and the observation by up to 5 K exist in the upper part of the sounding.

Examining the $V_H$–profiles, the IFS cycles simulate oscillations with vertical wavelength of $\lambda_z \approx 5 … 6$ km which are longer than the observed ones. Moreover, the amplitude of the resolved gravity waves is underestimated by the IFS. Nevertheless, the numerical results of the IFS suggest that the resolved gravity wave activity contains a fair degree of realism in the upper stratosphere confirming the findings of Dörnbrack et al. (2017a). Moreover, both the overall vertical change of wind and wind direction are well captured, especially by the IFS cycle 41r2. It should be noted that all the radiosonde profiles were not assimilated into the IFS and, therefore, constitute independent measurements.



*Inertia-Gravity Waves*

Figures 4 to 6 depict composites of operational analyses and the scale-dependent modal decomposition of the IFS cycle 41r2 at selected times on 30 January 2016. Figs. 4 and 5 juxtapose the magnitude of the IGW component of the horizontal wind $V_H^{IGW}$ (left column) and the horizontal divergence (DIV) patterns from the analyses (right column) at the upper stratospheric

pressure surfaces of 1, 3, and 5 hPa for 06 UTC and 12 UTC, respectively. The background fields in the individual panels of Figs. 4 and 5 are the geopotential height Z and the magnitude of the balanced wind $V_H^{BAL}$, respectively. Notice, both Z and DIV are extracted directly from the ECMWF data archive. Thus, the particular patterns of divergence and IGWs as computed by the modal decomposition can be considered as independent diagnostics of the unbalanced flow in the IFS.

Consecutive $V_H^{IGW}$-maxima mark groups of inertia-gravity waves of different orientations and intensities. There are two

groups over and in the lee of Scotland and Southern Scandinavia and another one over Northern Scandinavia, best seen in Fig. 4c. All groups reside along the inner edge of the polar vortex (Figs. 4a, c, e and 5a, c, e). The two groups over and in the lee of Scotland and Southern Scandinavia were likely excited by topographic forcing. This hypothesis is supported by the stationarity of the wave groups at different times, e.g. compare Fig. 4e and 5e. The $V_H^{IGW}$-maxima are correlated with undulations in the geopotential height Z at the different pressure levels. Over Scotland, the amplitude of $V_H^{IGW}$ increases with

height and does not change much from 06 to 12 UTC. The group over Southern Scandinavia weakens in time and splits in two parts at 12 UTC. The third wave group over Northern Scandinavia (Fig. 4c) is much weaker, i.e. reveals smaller $V_H^{IGW}$ amplitudes, compared to the others and, importantly, shows a transient behavior between 06 and 12 UTC. In the following, we concentrate on this wave group over Northern Scandinavia.

Two distinct features are relevant for the interpretation of the radiosonde observations. First, the inertia-gravity waves are

20 located in a region where the stratospheric jet is decelerated as visible by the declining wind $V_H^{BAL}$ towards northeast. There, the formation of the little vortex seen in $V_H^{BAL}$ in Figs. 4b and 5b leads to a broad divergent region over Northern Scandinavia. The gravity waves exist in a region where spontaneous adjustment likely excites inertia-gravity waves as known from studies of the tropospheric jet (e.g., Plougonven and Zhang, 2014).

Second, the southeastward displacement of the polar vortex causes a weakening of the horizontal wind above Northern

Scandinavia. On the other hand, the existence of the identified wave group is confined to regions with significant wind. Thus, the wave amplitude decreases in regions with weakening upper stratospheric winds $V_H^{BAL}$ (Figs. 5a, c, e). This becomes evident if one compares the $V_H^{IGW}$-amplitude at the different levels for 06 and 12 UTC. The northeast spreading of inertia-gravity wave activity at the inner edge of the polar vortex over Northern Scandinavia is due to the slight shift of the vortex position from 06 UTC to 12 UTC, cf. Figs. 4c, e with Figs. 5c, e and Figs. 4d, f with Figs. 5d, f.





Moreover, alternating positive and negative DIV-values indicate significant gravity wave activity located at the inner edge of the polar vortex. In general, the gravity wave activity is confined to regions where $V_H^{BAL}$ is larger than about 40 … 50 m s$^{-1}$. The threshold DIV values of $\pm\,2\cdot10^{-4}$ s$^{-1}$ are chosen as suggested by Dörnbrack et al. (2012) and as used by Khaykin et al. (2015) to locate hot spots of stratospheric gravity wave activity. In contrast to the $V_H^{IGW}$-field, the horizontal divergence is

not spectrally filtered and contains contributions from all resolved wavenumbers. Therefore, individual wave groups cannot easily be separated and the DIV patterns are not directly comparable to the $V_H^{IGW}$-field of the scale-dependent modal decomposition.

Figure 6 shows the temporal evolution of the zonal component $U^{IGW}$ and $V_H^{BAL}$ plotted as vertical cross-sections along the black line drawn in all panels of Figs. 4 and 5. The comparison of the three consecutive analysis times (00, 06, and 12 UTC)

clearly shows a gradual decrease of the stratospheric winds $V_H^{BAL}$ over Northern Scandinavia due to the southward displacement of the polar vortex during the minor SSW. At the northern (inner) edge of the polar vortex inclined phase lines are visible in the $U^{IGW}$-fields and indicate the gravity waves extracted from the normal modes analyses. At 00 UTC, gravity waves with a vertical wavelength of about $\lambda_z \approx 13$ km (estimated from the difference in Z between 3 and 30 hPa at 62°N) dominate the field in Southern Scandinavia (Fig. 6a). These waves are most likely related to the mountain waves visible in

the lower stratosphere between 60°N and 62°N. The increase of vertical wavelength with altitude is in accordance with linear wave theory as $\lambda_z \sim U/N$. Further south, the upper branch of the wave train excited over Scotland is seen above 10 hPa at 06 UTC (Fig. 6b). At the same time, a packet of gravity waves with $\lambda_z \approx 4\ …\ 5$ km is located at the polar-night jet's northernmost tip. It is this wave train we suggest to be excited by spontaneous adjustment in the divergent region of the PNJ. The altitude where the waves appear in the results of the modal decomposition is at about 10 hPa (~ 28 km). The vanishing

wave signature above about 3 hPa (~ 39 km) is consistent with the decreasing wind above this altitude (Fig. 2a, 5b, and 6b). At later times, the waves disappeared above Northern Scandinavia whereas only waves with longer vertical wavelengths seem to be trapped inside the PNJ between 20 hPa and 2 hPa (Fig. 6c). Altogether, the three snapshots indicate a highly transient wave progression at the inner edge of the polar vortex.

*Altitude-time sections*

Figure 7 presents the two quantities from the IFS simulations delineating gravity wave signatures as function of time and altitude over Kiruna, Sweden. Plots like Fig. 7 map the time-dependent 3D meteorological data into a 2D view as often provided by ground-based observations of vertical profilers (Dörnbrack et al., 2017b). Whereas the temperature fluctuations originate from a prognostic IFS variable, the vertical wind is a diagnostic quantity and it is used in a similar way as the horizontal divergence to visualize gravity waves. Between 26 and 29 January 2016, four sequences of vertically deep

propagating gravity waves appear as stacked positive and negative vertical velocity patterns extending to the upper stratosphere and mesosphere (Fig. 7a). Possible sources of these waves could be the weak cross mountain flow or the tropopause jet, cf. the times of the $V_H$-maxima in Fig. 2a with the appearance of the stratospheric gravity waves in Fig. 7a.



Afterwards, from 29 January noon to 30 January mid-night, the simulated stratospheric vertical wind patterns are not tied to the troposphere and tropopause region at this location. Furthermore, the smaller amplitudes suggest another excitation mechanism compared to the waves appearing in the preceding days. Apparently, the simulated waves disappeared above 50 km altitude concurrently with the ceasing horizontal wind after 29 January 00 UTC, cf. Fig. 2a. The nine balloon trajectories

as displayed in Fig. 7 indicate that only the deep sounding of the radiosonde launched in Kiruna on 30 January 2016 at 09:08 UTC partially penetrated the coherent stratospheric wave pattern above 25 km altitude near the end of the regime transition.

The temperature perturbations $T'$ were derived relative to an area mean as described in Section 2. The altitude-time section of $T'$ as shown in Fig. 7b delineates the very same gravity wave sequences as visible in vertical wind in Fig. 7a. Here, we relate the direction of the main axis of the elliptical patterns of positive or negative $T'$-values to phase lines indicating the

10 vertical direction of ground-based phase propagation. This is in contrast to altitude-distance plots, where the phase lines are normal to the direction of phase propagation (Sutherland, 2010, Fig. 1.25). For the sequences of gravity waves until 28 January 2016, the absolute $T'$-values increase up to 15 K in the upper stratosphere. The declining phase lines apparently suggest non-steady, upward propagating gravity waves. This first qualitative interpretation is based on linear wave theory where downward phase propagation is associated with upward energy propagation. However, the most striking feature of

15 Fig. 7b is the occurrence of ascending phase lines below about 35 km altitude during 29 January 2016. Linear wave theory suggests downward energy propagation for this period. The signature of this highly transient event fades on 30 January 2016. Afterwards, smaller-scale waves with lower amplitudes are observed following the morning of 31 January 2016 after the regime transition. In order to elucidate up- and downward propagating wave components quantitatively, we next apply the 2D wavelet analysis to the temperature fluctuations $T'$ shown in Fig. 7b.

*Wavelet Analysis of the Altitude-Time Sections of the IFS data above Kiruna*

2D wavelets as introduced in Section 2 are applied to quantify the stratospheric gravity wave activity and to deduce potential gravity wave sources as demonstrated by Kaifler et al. (2017). To classify the dominant vertical propagation directions of gravity waves, contributions from quasi-stationary waves (mountain waves, ground-based vertical phase speed $|c_{Pz}| \leq 0.036 \, \mathrm{m \, s^{-1}}$), upward propagating waves ($c_{Pz} < -0.036 \, \mathrm{m \, s^{-1}}$), and downward propagating waves ($c_{Pz} > 0.036 \, \mathrm{m \, s^{-1}}$) with

25 vertical wavelengths $\lambda_z$ between 2 and 15 km are separated and analyzed independently. Total potential energy densities $E_P^{tot}$ and $E_P$ -values for the three wave classes are derived and listed as averages over different altitude layers and for selected periods in Tables 1, 2, and 3. Averaging over the entire period from 26 January to 1 February, low $E_P^{tot}$-values of 1 and 3 J/kg are found below 28 km, compared to large values of 20 and 13 J/kg above 30 km altitude. Altogether, the dominant contributions above 30 km altitude are due to upward propagating gravity waves whereas contributions from quasi-stationary

waves are negligible (Table 3). On 29 January 2016, a strong enhancement of downward propagating waves is found in the whole column between 12 and 48 km altitude (see large $E_P^{down}$-values in Table 2). Below 38 km, $E_P^{down}$-values clearly surpass $E_P^{up}$-values and attain maximum values of 25 J/kg between 30 and 38 km altitude. Above 38 km altitude, $E_P^{up} >$





$E_P^{down}$ indicates wave generation at around that level in accordance with the ascending and descending phase lines in Fig. 7b. On 30 January 2016, upward propagating gravity waves dominate the $E_P$ -values at all altitude bins again. Note, the stratospheric values of $E_P^{tot}$ from the IFS agree quantitatively well with the $E_P$ -estimates from the radiosonde and reflect the same altitude dependence. For the IFS data, the ratio of upward propagating waves to all wave modes is calculated by

$R^{IFS} = E_P^{up}/(E_P^{mw} + E_P^{up} + E_P^{down})$ and $R^{IFS}$ is listed in Tables 1 to 3. A comparison with the radiosonde sounding data reveals both $R^{RS}$ and $R^{IFS}$-values larger than 0.6 for altitudes above 20 km indicating a dominance of upward propagating waves on 30 January 2016. In contrast, the strongly reduced $R^{IFS}$-values between 12 and 38 km on the day before clearly signify the dominance of downward propagating modes.

Temperature fluctuations $T'$ associated with up- and downward propagating waves as separated by the 2D wavelet analysis

are displayed as vertical time series in Fig. 8. Enhanced upper-stratospheric $T'$-values associated with upward propagating waves persist before and during the regime transition but their amplitude decreases on 30 January 2016 (Fig. 8a). Large amplitudes are reached during 28 January 2016 with vertical wavelengths $\lambda_z$ of 10 to 15 km. A similar pattern but of weaker amplitudes is found for downward propagating waves around the same time (Fig. 8b). However, a downward propagating wave packet dominates during the regime transition on 29 and 30 January 2016. It has significantly smaller vertical

wavelengths, a larger tilt, i.e. a higher phase speed, and is confined to the altitude range from 20 to about 45 km (Fig. 8b). The intermittent appearance of downward propagating stratospheric wave packets points to spontaneous adjustment as a possible generation mechanism in this short time period at the beginning and during the regime transition.

The distribution of gravity wave parameters is visualized in Fig. 9, where wavelet spectrograms are displayed as function of vertical wavelength $\lambda_z$, ground-based phase speed $c_{Pz}$, and altitude for four consecutive time periods. We show relative

wavelet spectrograms (see Sec. 2) in order to emphasize differences before, during and after the regime transition due to the minor SSW on 29 and 30 January 2016. The highest value zero means that the wave packet in question was detected in the selected interval only. The wave field is dominated by mostly upward-propagating waves with $|\lambda_z| \geq 10$ km and negative phase speeds $c_{Pz} \leq -0.05$ m s$^{-1}$ until 28 January 2016 (Fig. 9a, e). A slightly weaker wave with $|\lambda_z| \approx 8$ km and larger negative phase speed $c_{Pz} \leq -0.17$ m s$^{-1}$ is superimposed. The occurrence of downward propagating waves with $c_{Pz} \approx +$

0.17 m s$^{-1}$ and $\lambda_z \gtrsim 4$ km on 29 January 2016 is clearly visible in Fig. 9b. Fig. 9f shows their vertical extent between 12 and 40 km altitude. Simultaneously, upward-propagating waves with $\lambda_z = 5 \ldots 10$ km are found above 40 km altitude. This prominent dipole structure indicates a local gravity wave source at about 40 km altitude (Fig. 9f). The stratospheric gravity wave activity decreases dramatically on 30 and 31 January: A downward propagating wave packet of $\lambda_z \approx 5$ km and very small phase speed is detected in an altitude range between 40 and 48 km (Fig. 9c, g). A reconstruction of the downward

propagating wave packet with $\lambda_z = 4.7 \ldots 6.2$ km and with $c_{Pz} \approx +0.1$ m s$^{-1}$ is shown in Fig. 10. Finally, starting on 31 January, small-amplitude upward propagating waves with scales of $\lambda_z \approx 3$ km are found in the lower stratosphere at around 20 km altitude (Fig. 9d, h).





## 6. Discussion and Summary

In this paper, we analyzed upper stratospheric gravity waves which occurred at the inner edge of the Arctic polar vortex during a minor SSW end of January 2016. The gravity waves were observed by a singular radiosonde launched from Kiruna airport reaching an altitude of 38.1 km. This unusual peak altitude could be achieved, first, by using a large 3000 g balloon.

Second, the warming of the middle stratospheric cold layer from 180 K to 190 K helped maintaining the elasticity of the rubber skin until the balloon burst probably due to strong turbulence. This 10 K warming was associated with the southward displacement of the Arctic polar vortex. This means, the stratosphere above Kiruna was characterized by a distinct temporal change of the ambient flow conditions during the period from 29 to 30 January 2016: The warming and descent of the cold stratospheric layer went along with a gradual decrease of the horizontal winds during this regime transition from vortex edge

to center conditions (Fig. 2).

In agreement with previous observational studies (e.g., Whiteway et al., 1997, Yoshiki et al., 2004), significant stratospheric wave activity could be detected in the high-resolution IFS data at times when the PNJ was situated over Kiruna (cf. Figures 2a and 7a, b), i.e. before and during the regime transition. As shown by Le Pichon et al. (2015) or Ehard et al. (2018), IFS analyses are a reliable indicator of stratospheric gravity wave activity up to about 45 km altitude. Especially, the most recent

increase of the IFS horizontal resolution is able to reproduce realistic gravity wave amplitudes in the lower and middle stratosphere (Dörnbrack et al., 2017a). The IFS analyses and HRES short-term forecasts suggest that these stratospheric gravity waves were occurring whenever the tropopause jet was located over Kiruna (cf. Figs. 2a and 7a) in agreement with the findings of Whiteway and Duck (1999).

The observed values of $E_K$ and $E_P$ were large in the altitude region above about 30 km in the deep sounding of 30 January

2016 (Table 1). Compared to the earlier deep sounding on 29 January, the simultaneous decrease of upper tropospheric and lower stratospheric kinetic energies suggests a remote gravity wave source either in the troposphere or stratosphere. Except in the lower stratosphere from 12 to 20 km altitude, the dominant propagation direction of the analyzed gravity waves was upward ($R^{RS} > 0.6$) on 30 January 2016. A comparison of the two soundings from 29 to 30 January 2016 indicates a transition from downward to upward propagating waves in the middle stratosphere in the layer from 20 to 28 km altitude

($R^{RS}$ increased from 0.43 to 0.62, Tables 1 and 2). A value of $R^{RS} \sim 0.51$ between 12 and 20 km suggests a possible descent of the stratospheric layer dominated by downward propagating gravity waves from the middle to the lower stratosphere from 29 to 30 January.

The scale-dependent modal decomposition retrieved coherent wave packets of IGWs in the operational analyses of the IFS. Multiple stratospheric wave packets were identified over Northern Europe during the minor SSW. At least two of them are

related to orographic forcing over Scotland and Southern Scandinavia. The third wave packet over Northern Scandinavia seems to be generated by spontaneous adjustment of the transient stratospheric flow during the displacement of the Arctic



polar vortex and the deformation of the PNJ. The comparison with the horizontal divergence, which has traditionally been used as a proxy for IGW in the middle latitudes, demonstrates that the scale-dependent modal analysis can be successfully applied to diagnose the horizontal and vertical structure of inertia-gravity waves. A comparison with direct observations and their analysis by other methods suggests that the applied filtering method provides a useful way to quantify circulation variance associated with inertia-gravity waves in the IFS analyses. While the IFS analyses still lack small-scale variability associated with propagating IGWs, they resolve the main mesoscale features well in accordance with the findings of Žagar et al. (2017). Moreover, the separation of the quasi-geostrophic (i.e. balanced or Rossby wave) and IGW components of circulation revealed that a small-size vortex formed at the inner edge of the polar vortex generating a divergent stratospheric flow (Figs. 5 and 6) directly above Northern Scandinavia acting as a gravity wave source in a similar way as the exit region of the tropospheric jets (Plougonven and Zhang, 2014).

For the Northern Scandinavian wave packet, the normal mode analysis revealed inertia-gravity waves with $\lambda_H \approx 200 \ldots 300$ km propagating northeast. A similar horizontal wavelength ($\lambda_H \approx 220 \ldots 330$ km) and propagation direction was calculated from the Stokes analysis of the deep 30 January 2016 radiosonde sounding between 20 and 38 km altitude. In this altitude region, the small values of $\Omega/f \approx 3$ and $E_P \approx 3 \times E_K$ suggest inertia-gravity waves as the dominant mode in agreement with the normal mode analysis. The retrieved modes of scale-dependent modal decomposition obey the linear dispersion relation of inertia-gravity waves and confirm the observations of the radiosondes sounding. Moreover, the vertical wavelength $\lambda_z \approx 4$ km does not increase with height. Such an increase would be expected, however, for vertically propagating stationary hydrostatic gravity waves, i.e. mountain waves, due to increasing wind. Even an estimate of the vertical wavelength $\lambda_z = 2\pi\, U/N$ using $U \approx 40$ m s$^{-1}$ and $N \approx 0.025$ s$^{-1}$ results in a vertically longer wavelength of $\lambda_z \approx 10$ km. Thus, the observed rather small vertical wavelength supports the argument that the observed and simulated inertia-gravity waves origin from the same stratospheric source. The hypothesis of a stratospheric wave source is supported by the large values of $E_P$ and $E_K$ at the uppermost levels of the deep radiosonde profile as mentioned above. Furthermore, the existence of downward propagating waves is another indication for a stratospheric gravity source.

Downward propagating gravity waves were detected by the 2D wavelet analysis of vertical time sections of stratospheric $T'$-values over Kiruna during the regime transition (Fig. 10). In this way, the wavelet analysis also suggests an upper stratospheric gravity wave source as both the $E_P$–values for upward propagating waves in the layer above and for downward propagating waves in the layer below are significantly increased (Tables 1 and 2). In this analysis, negligible $E_P$–values associated with quasi-stationary gravity waves most likely exclude a reflection of mountain waves as cause of the downwards propagating waves. In order to substantiate our analysis, we calculated the temperature fluctuations $T'$ also as difference of the fully resolved IFS fields (using 1279 spectral coefficients) minus the IFS fields retrieved for horizontal wavenumbers smaller than 21 and 42. Latter data constitute the modified background fields. The partitioning of potential



energies as listed in Tables 1 to 3 does not change considerably due to the different T′-estimates (not shown) and reveals the robustness of the applied methodology.

A critical point of the analysis of vertical time series is the Doppler shift which might swap the orientation of phase lines leading to a false assignment of vertical energy propagation (e.g, Kaifler et al., 2017, Dörnbrack et al., 2017b). Here, we

provide an independent argument that downward propagating waves can be detected in the IFS analyses and HRES short-term forecasts. For this purpose, perturbation pressure and vertical wind were calculated with the same method as mentioned above for T′. Stratospheric vertical energy fluxes $p'w'$ were computed as temporal and horizontal averages for an area around Kiruna and the results are listed for selected stratospheric altitude in Table 4. Negative energy fluxes occur between 20 and 38 km altitude on 29 January and at lower levels at around 06 UTC on 30 January. It must be noted that negative

fluxes only appeared in this area around and northeast of Kiruna. An analysis of hourly fluxes for the large area as used to determine the background profiles (see Sec. 2) never resulted in negative energy fluxes. Therefore, the spatially confined and intermittent appearance of downward propagating gravity waves is suggestive of their transient and short-lived nature.

Usually, stratospheric gravity wave activity in the Arctic is highly correlated with the surface winds (e.g., Yoshiko and Sato, 2000). Nevertheless, the combination of observational and numerical data and the application of different analysis methods

provide evidence of the existence of an elevated source of gravity wave excitation sporadically occurring during disturbances of the polar vortex by planetary waves, e.g. during SSWs.

**Acknowledgements**

Part of this research was supported by the German research initiative "Role of the Middle Atmosphere in Climate (ROMIC/01LG1206A)" funded by the German Ministry of Research and Education in the project "Investigation of the life

cycle of gravity waves (GW-LCYCLE/01LG1206A)" and "Mesoscale Processes in Troposphere-Stratosphere Interaction" (METROSI/01LG1218A). Furthermore, the Deutsche Forschungsgemeinschaft (DFG) supported SG, MB, and TP in the Project "Multiscale Dynamics of Gravity Waves" (MS-GWaves with the subprojects GW-TP/DO 1020/9-1, PACOG/RA 1400/6-1). Nedjeljka Žagar and Damjan Jelić were funded by the European Research Council (ERC), Grant Agreement no. 280153, MODES. Access to the ECMWF data was possible through the special project "HALO Mission Support System".

The technical support by Maria Siller, Andreas Schneider, and Michael Priester is greatly appreciated.



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



**Tables**

**Table 1**: Kinetic and potential energy densities $E_K$ ($m^2$ $s^{-2}$) and $E_P$ (J $kg^{-1}$) for the 30 January 2016 09:08 UTC radiosonde soundings (RS). Additionally, the ratio $R^{RS}$ of the power of the upward propagating part of the rotary spectrum to the total power is given. Mean gravity wave potential energy densities in J $kg^{-1}$ derived from vertical time series of the IFS

5 temperature perturbations as shown in Fig. 7b. $E_P$–values are classified for quasi-stationary mountain waves ($E_P^{mw}$, $|c_{Pz}|$ ≤ 0.036 m $s^{-1}$), upward propagating waves ($E_P^{up}$, $c_{Pz}$ < 0.036 m $s^{-1}$), downward propagating waves ($E_P^{down}$, $c_{Pz}$ > 0.036 m $s^{-1}$), and the total value $E_P^{tot}$ from the 2D wavelet analysis. Due to the non-linear average of $T'$ in the calculation of the gravity wave potential energy densities, the sum $E_P^{mw} + E_P^{up} + E_P^{down}$ deviates slightly from $E_P^{tot}$. For the IFS analysis $R^{IFS}$ is calculated as ratio $E_P^{up}/(E_P^{mw} + E_P^{up} + E_P^{down})$.

| | 30 January 2016 | | | | | | | |
| --- | --- | --- | --- | --- | --- | --- | --- | --- |
| | RS | | | IFS | | | | |
| | $E_P$ | $E_K$ | $R^{RS}$ | $E_P^{mw}$ | $E_P^{up}$ | $E_P^{down}$ | $R^{IFS}$ | $E_P^{tot}$ |
| 1–8 km | 10 | 8 | 0.78 | | | | | |
| 12–20 km | 5 | 10 | 0.51 | 0 | 1 | 0 | 0.60 | 1 |
| 20–28 km | 9 | 23 | 0.62 | 0 | 2 | 0 | 0.68 | 3 |
| 30–38 km | 25 | 79 | 0.76 | 3 | 14 | 5 | 0.66 | 29 |
| 40–48 km | | | | 0 | 6 | 3 | 0.65 | 11 |



**Table 2**: As Table 1 for the 29 January 2016 10:42 UTC sounding and IFS temperature perturbation on 29 January 2016.

| | 29 January 2016 | | | | | | | |
| --- | --- | --- | --- | --- | --- | --- | --- | --- |
| | RS | | | IFS | | | | |
| | $E_P$ | $E_K$ | $R^{RS}$ | $E_P^{mw}$ | $E_P^{up}$ | $E_P^{down}$ | $R^{IFS}$ | $E_P^{tot}$ |
| 1–8 km | 5 | 16 | 0.74 | | | | | |
| 12–20 km | 3 | 12 | 0.47 | 0 | 0 | 2 | 0.21 | 2 |
| 20–28 km | 5 | 21 | 0.43 | 0 | 1 | 5 | 0.17 | 7 |
| 30–38 km | | | | 1 | 8 | 25 | 0.24 | 39 |
| 40–48 km | | | | 0 | 15 | 6 | 0.72 | 21 |

**Table 3**: IFS results as in Tables 1 and 2 for the period from 26 January until 1 February 2016.

| | 26 January – 1 February 2016 | | | | |
| --- | --- | --- | --- | --- | --- |
| | IFS | | | | |
| | $E_P^{mw}$ | $E_P^{up}$ | $E_P^{down}$ | $R^{IFS}$ | $E_P^{tot}$ |
| 1–8 km | | | | | |
| 12–20 km | 0 | 1 | 1 | 0.45 | 1 |
| 20–28 km | 0 | 1 | 1 | 0.50 | 3 |
| 30–38 km | 2 | 10 | 6 | 0.57 | 20 |
| 40–48 km | 0 | 9 | 2 | 0.76 | 13 |

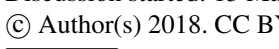



**Table 4**: Stratospheric vertical energy fluxes $p'w'$ in W/m$^2$ from the IFS HRES analyses and short-term forecasts averaged over the area 68°N to 69°N and 18°E to 22°E over the respective times and altitude bins.

|  | 29 Jan 11-13 UTC | 29 Jan 17-19 UTC | 29/30 Jan 23-01 UTC | 30 Jan 05-09 UTC | 30 Jan 11-13 UTC |
|---|---|---|---|---|---|
| 12 - 20 km | 0.511 | 0.118 | 0.380 | - 0.287 | 0.218 |
| 20 - 28 km | 0.010 | 0.371 | - 0.111 | 0.226 | 0.161 |
| 30 - 38 km | - 0.046 | 0.095 | - 0.066 | 0.198 | 0.088 |
| 40 - 48 km | 0.098 | 0.023 | 0.015 | 0.075 | 0.082 |







**Figure 1**: Magnitude of the balanced wind $V_H^{BAL}$ (m s$^{-1}$, color shaded) from the normal mode analysis at pressure surfaces of 1 hPa (a), 5 hPa (b), 50 hPa (c), and 250 hPa (d) valid on 30 January 2016 06 UTC.



**Figure 2**: Altitude-time sections of the horizontal wind $V_H$ in m s$^{-1}$ (a) and the absolute temperature in K (b) as a composite of 1 hourly short-term HRES forecasts and 6 hourly operational analyses from the IFS cycle 41r2 above Kiruna, Sweden. The thin black lines are the logarithm of the potential temperature with constant increments of 0.05. The vertical black lines mark the radiosondes paths of the nine soundings mentioned in the text.





**Figure 3**: Vertical profiles of the absolute temperature (a), potential temperature (b), buoyancy frequency (c), horizontal wind (d), wind direction (e), and vertical wind (f). Red dots: radiosonde observations from the 30 January 2016 0908 UTC ascent. Solid lines: ECMWF IFS operational HRES forecasts interpolated in space and time on the balloon trajectory from cycle 41r1 (black) and 41r2 (blue). The ECMWF vertical winds are multiplied by a factor 10. The vertical winds from the balloon sounding are calculated as difference of local ascent rate and a mean ascent rate of 6.1 m/s determined as integral value over the whole profile.



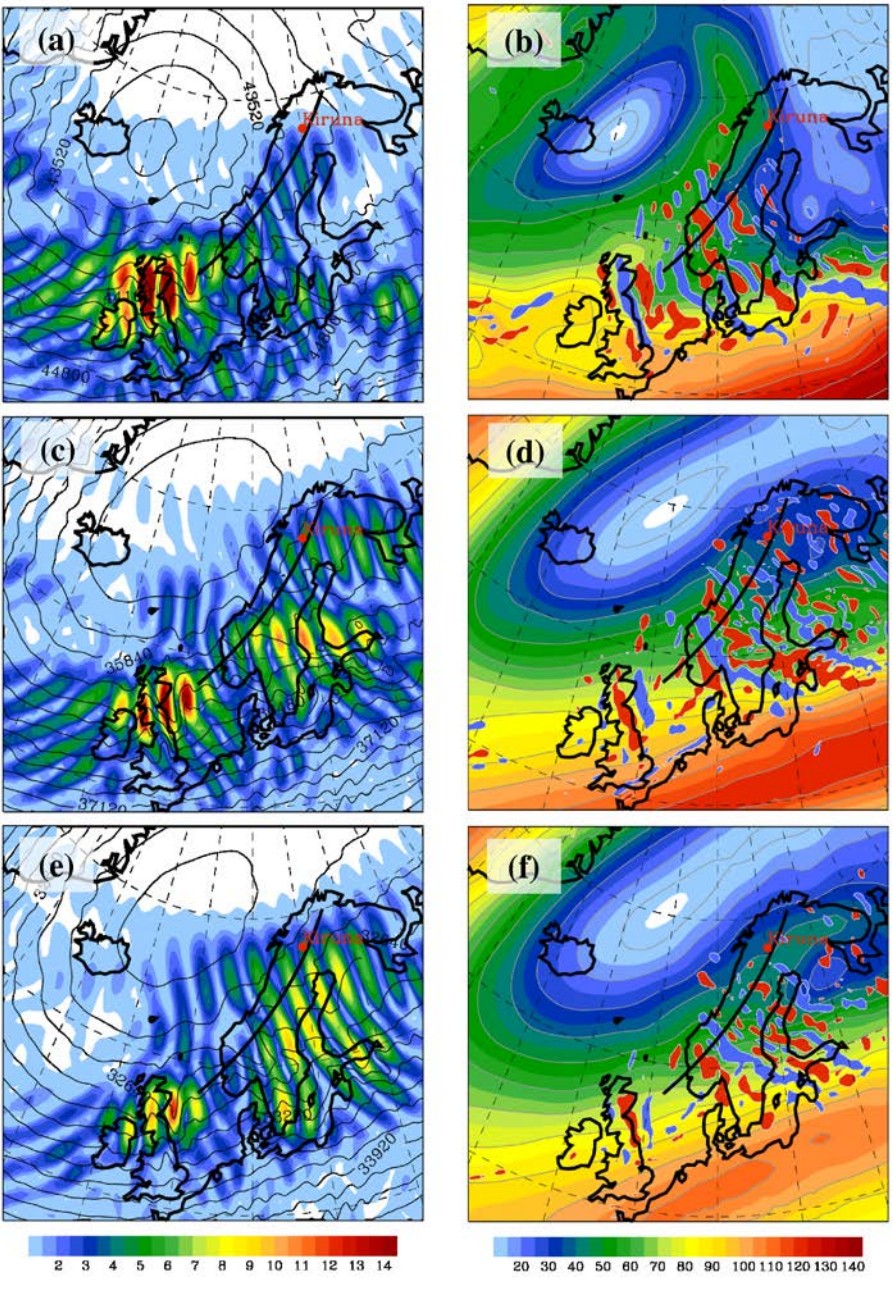

**Figure 4**: Left column: composite of the magnitude of $V_H^{IGW}$ (m s$^{-1}$, color shaded) from the normal mode analysis and the geopotential height (m, black contour lines) from IFS operational analyses. Right column: composite of the magnitude of the balanced wind $V_H^{BAL}$ (m s$^{-1}$, color shaded) from the normal mode analysis and the horizontal divergence (values larger/smaller ± 2·10$^{-4}$ s$^{-1}$ are filled with red and and blue, respectively) from operational analyses. The plots are at 1 hPa (~ 48 km; a, b), 3 hPa (~ 40 km; c,d), and 5 hPa (~ 36 km; e, f) and they are valid at 06 UTC on 30 January 2016. The black line marks the baseline of the vertical sections shown in Fig. 6.



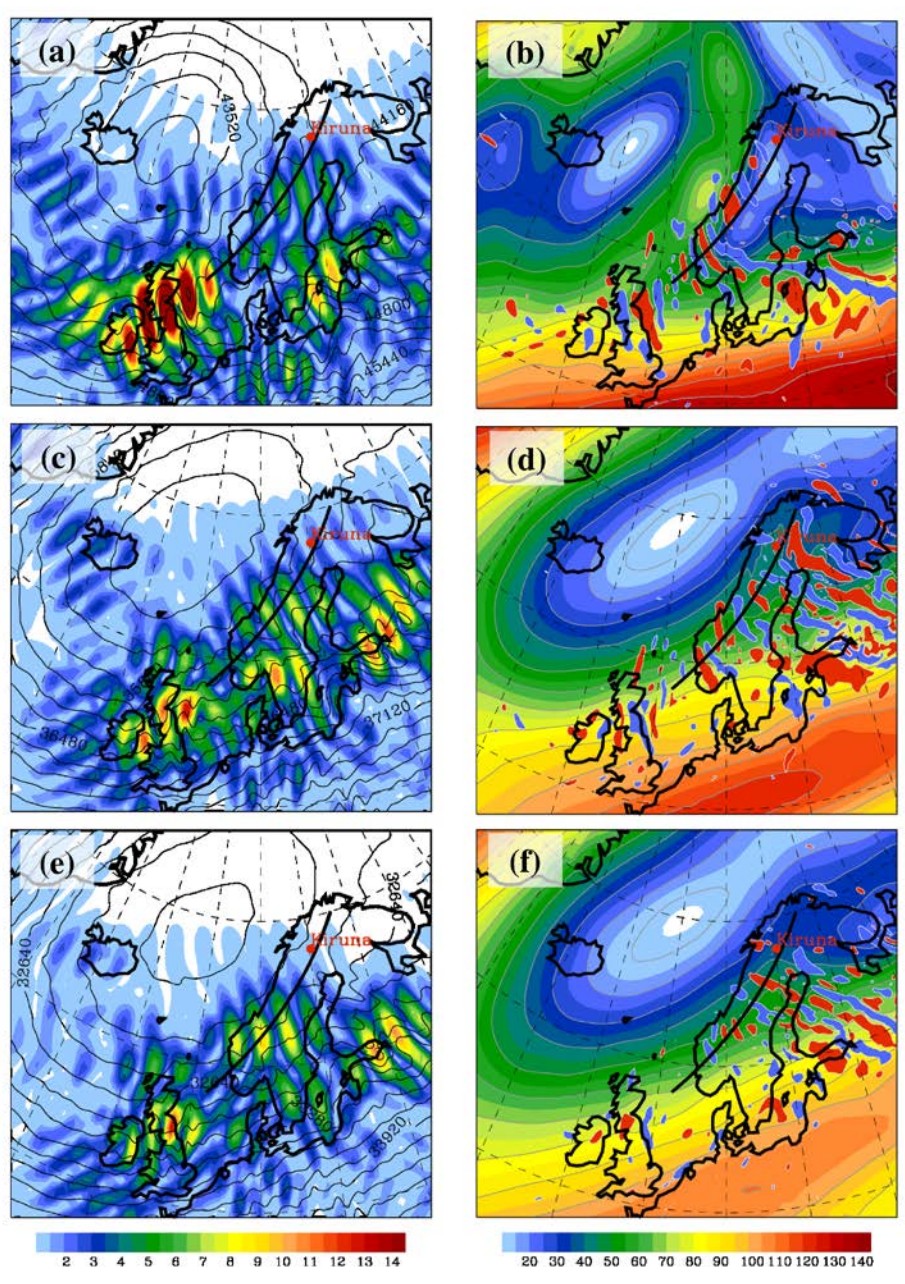

**Figure 5**: Same as Fig. 4 for 12 UTC on 30 January 2016.





**Figure 6**: Composite of the magnitude of the balanced wind $V_H^{BAL}$ (m s$^{-1}$, color shaded) and the unbalanced zonal wind $U^{IGW}$ (areas with negative values larger -3 m s$^{-1}$ and positive values smaller 3 m s$^{-1}$ are filled with blue and red, respectively) from the normal mode analysis. The vertical sections are along the baseline sketched in Figs. 4 and 5 and they are valid on 30 January 2016 at 00 UTC (a), 06 UTC (b), and 12 UTC (c).



**Figure 7**: Altitude-time sections of the vertical wind w in m s$^{-1}$ (a, $\Delta w = 0.1$ m s$^{-1}$) and temperature perturbations T' in K (b, $\Delta T' = 2$ K) as a composite of 1 hourly short-term HRES forecasts and 6 hourly operational analyses from the IFS cycle 41r2 above Kiruna, Sweden. Positive and negative values are plotted with red and blue lines. The temperature fluctuations are calculated relative to an area mean. In panel (a), minimum and maximum w-values are -1.5 m s$^{-1}$ and 1.2 m s$^{-1}$ and in panel (b) minimum and maximum T'-values are ± 15 K, respectively. The thin black lines are the logarithm of the potential temperature with constant increments of 0.05. The vertical black lines mark the radiosondes paths of the nine soundings mentioned in the text.



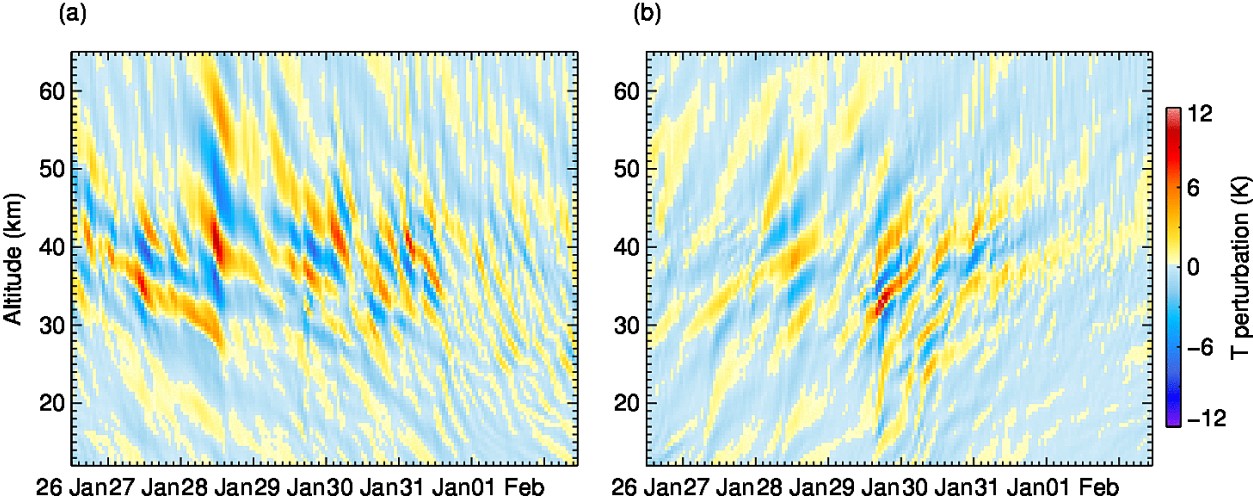

**Figure 8**: Temperature fluctuations associated with (a) upward and (b) downward propagating gravity waves as reconstructed by wavelet analysis (see text).





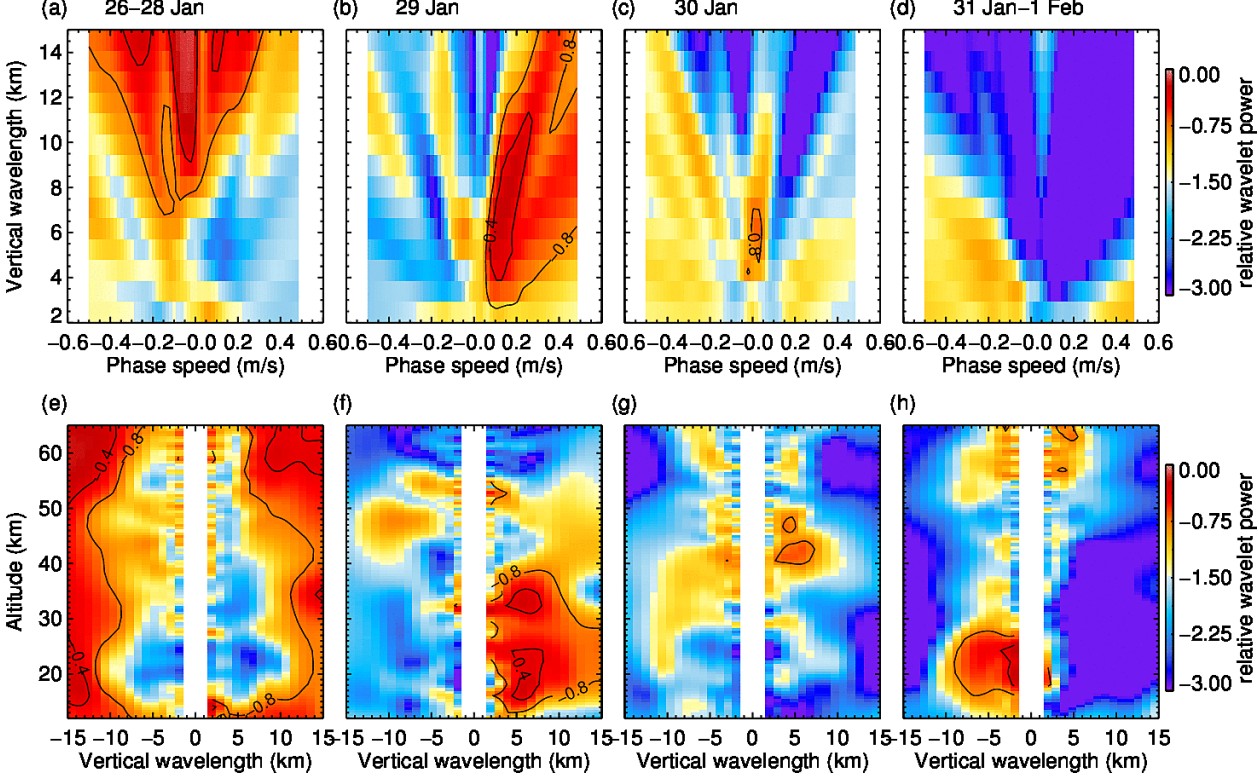

**Figure 9**: Relative wavelet power as function of vertical wavelength $\lambda_z$ and vertical phase speed $c_{Pz}$ with respect to the background spectrogram for four time periods.



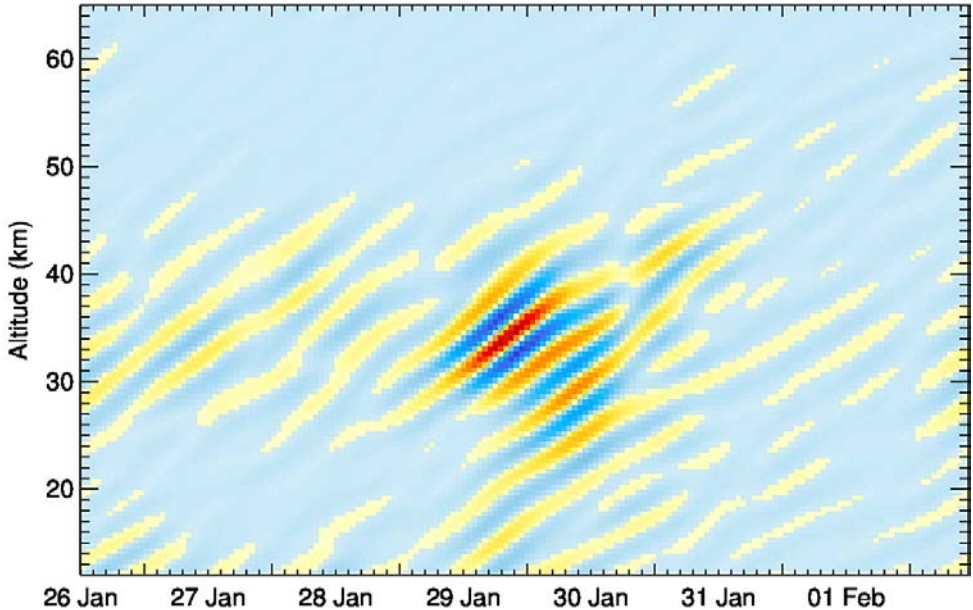

**Figure 10**: Downward propagating waves with $\lambda_z = 4.7 \dots 6.2$ km and $c_{Pz} \approx 0.1$ m s$^{-1}$ occurring during 29 to 30 January
2016 as reconstructed by wavelet analysis.