# Peer review of "Gravity Waves excited during a Minor Sudden Stratospheric Warming"

_Atmospheric Chemistry and Physics, 2018_

## Referee Comment (RC1) · Anonymous Referee #1 · 7 Apr 2018

The paper presents strong observational evidence for inertia-gravity wave generation during an Arctic winter minor stratospheric sudden warming. By performing a detailed analysis of both an exceptionally deep radiosonde sounding over Northern Scandinavia and ECMWF analysis/forecast, the study highlights the role of internal stratospheric dynamics as a source of gravity wave generation. The novel aspect of the paper is illustration that spontaneous generation of inertia-gravity waves can occur from balanced dynamics in the stratosphere. Spontaneous emission from deformation of stratospheric polar vortex has thus far received little attention, as most spontaneous gravity wave emission studies have focused on jet exit regions and surface fronts in the troposphere. I find the paper to be very well written and motivated and therefore highly recommend it for publication in its present form.

---

## Referee Comment (RC3) · Anonymous Referee #3 · 8 May 2018

This is an interesting manuscript providing several indications for an observation of the excitation of internal gravity waves by the spontaneous adjustment of a sub-vortex at the inner edge of the Arctic polar vortex during a minor sudden stratospheric warming. Unlike most scenarios considered to date, where the spontaneous excitation occurs in the troposphere, e.g. in jet-exit regions, here this process seems to be at work in the stratosphere. The authors make their case by an impressive combination of analyses, both of observational radiosonde data and of IFS simulation data from ECMWF. Several strong indicators are presented, e.g. upward/downward energy fluxes above/below the presumed emission region, corresponding downward/upward phase propagation, Stokes analyses applicable to nearly monochromatic wave fields, a configuration of the balanced flow (diagnosed by a modal analysis) conducive to spontaneous emission in a

way similar to typical tropospheric situations, and wavepacket reconstructions verifying the assumed vertical propagation directions. I therefore recommend acceptance of the manuscript, provided my comments below are taken into account.

Main Review Points

1. As tempting as the conclusion on spontaneous emission seems to be, I am not (yet) convinced completely:

a. The dynamical situation is discussed using to a large part Figs. 4 and 5, that show a complex situation with several wave packets due to different origins. The focus is on a comparatively weak signal over Northern Scandinavia that is co-located with a sub-vortex formed at the edge of the polar vortex. As I understand the authors, they base their assumption of spontaneous adjustment on the jet deceleration observed there. This might be correct, but appears somewhat speculative nonetheless. It might be impossible to provide a real proof, but the authors could collect more evidence for the balanced flow getting out of balance. A standard indicator is the local Rossby number, e.g. determined from the ratio between relative and planetary vorticity, that should be larger in regions of spontaneous emission.

b. Based on low potential-energy values, the authors argue in the conclusions against the reflection of mountain waves. This sounds reasonable, but can one exclude that a low-frequency gravity wave, approaching the jet obliquely from below, is reflected by the latter? How can one make sure that such a reflection process is not taking place at some horizontal distance, and that one locally obtains a situation that looks like emission but really is refraction by the jet? I would do not have a problem if the basis of evidence remained as it is now, but then I would recommend formulating abstract and conclusions even more openly, and maybe also changing the title to 'Indications for . . .', e.g.

2. The interesting modal analysis, used for separating balanced flow from gravity waves, could be exploited even more:

a. If (!) the authors were interested I would find it really useful to see the wavelet analysis of the IFS data redone, most importantly Fig. 9, for the IGW part only. As in many other publications the authors directly identify the deviations from some mean with gravity waves. This assumes (!) that balanced motions do not contribute significantly to the small scales, but here would be a chance to demonstrate this for the given case.

b. Also, in an imbalance analysis using local Rossby numbers, how much do the mesoscale balanced motions contribute to the result?

Minor review points

1. p. 4, l. 28: "semi-implicit model" does not sound correct. It is just the time step that is semi-implicit.

2. p. 5, section "Scale-dependent . . .": I sympathize with the linear modal analysis, yet it is linear and a short comment might be in place, why no higher-order, nonlinear, balance concept is used, as are discussed by Plougonven and Zhang (2014), e.g.

3. p. 6, ls. 14 - 19: Kaifler et al (2015, 2017) do not use quite the same relative spectrograms. They subtract the average instead of the sum. Would be better if this were described correctly.

4. ps. 8 – 10, section "Gravity wave analysis": Here and at many other places local fluctuations are identified with gravity waves, and additional analysis is done to demonstrate the consistency of this assumption. Often this is unavoidable, but here modal analysis of IFS data could be used to support this hypothesis even more.

5. p. 9, ls. 8: Rather "dominated by" than "have"? Some of the smaller-scale fluctuations, that IFS does not simulate, might have larger frequencies.

6. p. 11, ls. 7 - 8: I do not quite agree that modal decomposition and horizontal divergence are independent diagnostics. By its polarization relations, being geostrophic, extratropical balanced flow, as diagnosed from the modal analysis, is non-divergent.

Moreover, both analyses might incorrectly attribute divergent flow to gravity waves, while it might as well belong in parts to the balanced flow, as e.g. can be diagnosed from the omega equation and other higher-order balance concepts. I agree that it is meaningful to investigate horizontal divergence, but one should also admit possible issues . . .

7. ps. 11 – 12, section "Gravity waves": This is the section that I find somewhat speculative, see my major comment 1.

8. ps. 13 – 14, section "Wavelet Analysis . . .", esp. Fig. 9: Here the assumption is used that vertical gravity-wave phase velocities oppose the corresponding group velocities. Fortunately, in the conclusions the authors mention the risk of misinterpretations due to the Doppler term, and invoke energy flux as well. However, in the present case where all information is available in the IFS data, would it not be better to avoid this risky analysis, and rather show group velocities directly?

---

## Author Comment (AC1) · 23 Jul 2018

Answers to the reviews of the paper

"Gravity Waves excited during a Minor Sudden Stratospheric Warming"

submitted to the journal Atmospheric Physics and Chemistry. In the following, we reply to the remarks point-by-point. The text of the reviewers is in black, our responses are written in red.

**Reviewer #1**

The paper presents strong observational evidence for inertia-gravity wave generation during an Arctic winter minor stratospheric sudden warming. By performing a detailed analysis of both an exceptionally deep radiosonde sounding over Northern Scandinavia and ECMWF analysis/forecast, the study highlights the role of internal stratospheric dynamics as a source of gravity wave generation. The novel aspect of the paper is illustration that spontaneous generation of inertia-gravity waves can occur from balanced dynamics in the stratosphere. Spontaneous emission from deformation of stratospheric polar vortex has thus far received little attention, as most spontaneous gravity wave emission studies have focused on jet exit regions and surface fronts in the troposphere. I find the paper to be very well written and motivated and therefore highly recommend it for publication in its present form.

**Reviewer #2**

"The paper analyzes stratospheric gravity waves that occurred at the edge of the polar vortex during a minor SSW. Although multiple stratospheric packets are discussed, of particular importance is a suggestion of a divergent stratospheric flow acting as a gravity wave source. As it is a matter of a debate whether gravity waves can be generated in the stratosphere, study supporting stratospheric wave source represents a substantial contribution to scientific progress. The scientific approach and applied methods are valid and thoroughly discussed. The paper properly discusses results of this study in the context of current understanding, and gives sufficient consideration and credit to related work. The paper is well written, concise and clear. I recommend to publish this paper.

We appreciate the extremely positive responses of both reviewers to our paper. Thank you very much!

**Reviewer #3**

This is an interesting manuscript providing several indications for an observation of the excitation of internal gravity waves by the spontaneous adjustment of a sub-vortex at the inner edge of the Arctic polar vortex during a minor sudden stratospheric warming. Unlike most scenarios considered to date, where the spontaneous excitation occurs in the troposphere, e.g. in jet-exit regions, here this process seems to be at work in the stratosphere. The authors make their case by an impressive combination of analyses, both of observational radiosonde data and of IFS simulation data from ECMWF. Several strong indicators are presented, e.g. upward/downward energy fluxes above/below the presumed emission region, corresponding downward/upward phase propagation, Stokes analyses applicable to nearly monochromatic wave fields, a configuration of the balanced flow (diagnosed by a modal analysis) conducive to spontaneous emission in a way similar to typical tropospheric situations, and wavepacket reconstructions verifying the assumed vertical propagation directions. I therefore recommend acceptance of the manuscript, provided my comments below are taken into account.

We thank referee #3 for the positive feedback.

*Main Review Points*

1. As tempting as the conclusion on spontaneous emission seems to be, I am not (yet) convinced completely:

a. The dynamical situation is discussed using to a large part Figs. 4 and 5, that show a complex situation with several wave packets due to different origins. The focus is on a comparatively weak signal over Northern Scandinavia that is co-located with a subvortex formed at the edge of the polar vortex. As I understand the authors, they base their assumption of spontaneous adjustment on the jet deceleration observed there. This might be correct, but appears somewhat speculative nonetheless. It might be impossible to provide a real proof, but the authors could collect more evidence for the balanced flow getting out of balance. A standard indicator is the local Rossby number, e.g. determined from the ratio between relative and planetary vorticity, that should be larger in regions of spontaneous emission.

In this paper, we present only two physical variables to quantify unbalanced motion components of the total flow. First, we showed the horizontal divergence – a commonly used diagnostics – and, secondly, we used the modal decomposition to differentiate between balanced and unbalanced modes. Both fields are juxtaposed in Figures 4 and 5 for two different times. The use of the unbalanced wind $V^{IGW}$ is a useful, complementary, and novel tool for identifying regions of spontaneous emission of inertia-gravity waves.

Nevertheless, and as suggested by the referee #3, we calculated the ratio of relative to planetary vorticity. The results are shown in the following figures at two different pressure levels (10 hPa and 300 hPa) for 30 January 2016 06 and 12 UTC, respectively.

[Figure]

Fig. 1: Ratio of relative to planetary vorticity at 10 hPa (left panels) and at 300 hPa (right panels) for 30 January 2016 at 06 UTC (top row) and at 12 UTC (bottom row).

At the 10 hPa pressure level, the positive values reveal enhanced local ratios over Scandinavia that propagate northwards and which are related to the PNJ. At the lower 300 hPa level, regions of increased values are located to the east of Kiruna and over Northern Scotland, respectively. According to the request of referee #3, this additional diagnostics confirms our conclusions about the stratospheric jet likely being the source of the observed inertia-gravity waves. It must be noted that the stratospheric patterns as displayed in Fig. 1 are temporally limited to the early 12 h of 30 January 2016. They never occur before and after this period, not shown.

b. Based on low potential-energy values, the authors argue in the conclusions against the reflection of mountain waves. This sounds reasonable, but can one exclude that a low-frequency gravity wave, approaching the jet obliquely from below, is reflected by the latter? How can one make sure that such a reflection process is not taking place at some horizontal distance, and that one locally obtains a situation that looks like emission but really is refraction by the jet? I would do not have a problem if the basis of evidence remained as it is now, but then I would recommend formulating abstract and conclusions even more openly, and maybe also changing the title to 'Indications for . . .', e.g.

We do not see any indication of the existence of critical levels for vertically propagating mountain waves as the mean flow velocity is always larger than 0. Therefore, the intrinsic frequency for mountain waves $\Omega = -kU - lV$ never becomes zero in the troposphere and stratosphere. Inside the jet, the intrinsic frequency $\Omega \to N$ which causes turning levels where the vertical wave number $m$ becomes infinity. In these regions indeed, gravity waves are reflected. A rough estimate of the vertically propagating wave modes based on the Scorer parameter shows that horizontal wavelengths larger than about 40 km can propagate freely into the stratosphere and we see no indication of wave reflection of longer waves in our case. This is also in agreement with results based on linear wave theory by Schoeberl (1985) for generic profiles for winter and equinox winds[1]. Because the wavelet analysis confirms these simple considerations, we do not intent to change the title of the paper.

2. The interesting modal analysis, used for separating balanced flow from gravity waves, could be exploited even more:

a. If (!) the authors were interested I would find it really useful to see the wavelet analysis of the IFS data redone, most importantly Fig. 9, for the IGW part only. As in many other publications the authors directly identify the deviations from some mean with gravity waves. This assumes (!) that balanced motions do not contribute significantly to the small scales, but here would be a chance to demonstrate this for the given case.

We agree that the modal decomposition should be exploited more extensively. We postpone this task for future projects. So far, the results of the modal analysis are only available in six-hourly intervals. To produce plots for a wavelet analysis similar to Fig. 9, one-hourly output were necessary. We plan t to do this in future applications using results of the modal analysis.

b. Also, in an imbalance analysis using local Rossby numbers, how much do the mesoscale balanced motions contribute to the result?

To answer this question, more studies with the proposed diagnostics were necessary. The results shown in Fig. 1 are obtained for IFS fields retrieved with 21 spectral coefficients. One could continue to analyze fields with larger zonal wavenumbers extending well into the mesoscale. However, this is beyond the scope – and intention – of our paper.
* * *
[1] Schoeberl, M.R., 1985: The Penetration of Mountain Waves into the Middle Atmosphere. J. Atmos. Sci., 42, 2856–2864.

*Minor review points*

1. p. 4, l. 28: "semi-implicit model" does not sound correct. It is just the time step that is semi-implicit.

We changed the respective sentence into:

"The IFS is a global, hydrostatic NWP model with semi-implicit time stepping (Robert et al., 1972) and semi-Lagrangian advection (Ritchie, 1988)."

The respective quotations were added to the list of references.

2. p. 5, section "Scale-dependent . . .": I sympathize with the linear modal analysis, yet it is linear and a short comment might be in place, why no higher-order, nonlinear, balance concept is used, as are discussed by Plougonven and Zhang (2014), e.g.

It is not common in journals to specify all possible approaches which potentially could be used. We did clearly say what we do and this should be sufficient. Knowledgeable scientists like you, know about other approaches, other people might be confused and become distracted. So, we decided to not add another paragraph. Otherwise, the implications were manifold: one had to start defining terms correctly, one had to compare and to evaluate the different approaches, and one had to defend the decision of a specific diagnostics in a way that all possible choices had the same practical applicability. This is by far not true. In reality, one uses the tools one has the command of and that one wants to explore more deeply.

3. p. 6, ls. 14 - 19: Kaifler et al (2015, 2017) do not use quite the same relative spectrograms. They subtract the average instead of the sum. Would be better if this were described correctly.

We have changed the respective sentences to clarify our approach:

"…of dominant gravity waves are estimated. Recently, this technique was developed and applied successfully to time series of ground-based Rayleigh lidar profiles. It is described and discussed in detail by Kaifler et al. (2015, 2017). To investigate the evolution of the gravity wave field around 30 January relative spectrograms are determined. While Kaifler et al. (2017) applied a seasonal average for comparison, here, we use the global spectrogram computed over the whole period from 26 January until 1 February 2016 as a reference. The relative spectrogram then is the difference between the spectrogram computed for selected intervals and this global spectrogram. As the latter contains the sum …"

4. ps. 8 – 10, section "Gravity wave analysis": Here and at many other places local fluctuations are identified with gravity waves, and additional analysis is done to demonstrate the consistency of this assumption. Often this is unavoidable, but here modal analysis of IFS data could be used to support this hypothesis even more.

See the point 2a about the modal analysis above.

5. p. 9, ls. 8: Rather "dominated by" than "have"? Some of the smaller-scale fluctuations, that IFS does not simulate, might have larger frequencies.

Correct! The sentence was changed accordingly.

6. p. 11, ls. 7 - 8: I do not quite agree that modal decomposition and horizontal divergence are independent diagnostics. By its polarization relations, being geostrophic, extratropical balanced flow, as diagnosed from the modal analysis, is non-divergent.

First of all, horizontal divergence is one of the prognostic variables of the IFS. Therefore, the balanced part of the modal decomposition comes from the horizontal wind components and constitutes an independent diagnostics. Moreover, the modal analysis shows that there is little divergence in balanced flow.

Moreover, both analyses might incorrectly attribute divergent flow to gravity waves, while it might as well belong in parts to the balanced flow, as e.g. can be diagnosed from the omega equation and other higher-order balance concepts. I agree that it is meaningful to investigate horizontal divergence, but one should also admit possible issues . . .

Exactly, this was our point: To avoid misinterpretation of divergent modes as gravity waves, we use the modal decomposition where we know that the IGW-part satisfies the dispersion relation!

7. ps. 11 – 12, section "Gravity waves": This is the section that I find somewhat speculative, see my major comment 1.

See our reply above.

8. ps. 13 – 14, section "Wavelet Analysis . . .", esp. Fig. 9: Here the assumption is used that vertical gravity-wave phase velocities oppose the corresponding group velocities. Fortunately, in the conclusions the authors mention the risk of misinterpretations due to the Doppler term, and invoke energy flux as well. However, in the present case where all information is available in the IFS data, would it not be better to avoid this risky analysis, and rather show group velocities directly?

The computation of the vertical energy flux based on the IFS data was indeed motivated by the idea to provide an independent diagnostics to determine the sign of the vertical energy propagation. In contrast to the comment of referee #3, we do not consider this application of an exact mathematical definition as " … risky analysis, … ". As far as we know, group velocities can be determined by conducting ray tracing calculations as done by Preusse et al. (2014)[2]. We leave this task for future studies.
* * *
[2] Preusse, P., Ern, M., Bechtold, P., Eckermann, S. D., Kalisch, S., Trinh, Q. T., and Riese, M.: Characteristics of gravity waves resolved by ECMWF, Atmos. Chem. Phys., 14, 10483-10508, https://doi.org/10.5194/acp-14-10483-2014, 2014.